# A novel virtual screening procedure identifies Pralatrexate as inhibitor of SARS-CoV-2 RdRp and it reduces viral replication *in vitro*

**Haiping Zhang**[1☯], **Yang Yang**[2☯], **Junxin Li**[3☯], **Min Wang**[4], **Konda Mani Saravanan**[1], **Jinli Wei**[2], **Justin Tze-Yang Ng**[5], **Md. Tofazzal Hossain**[1,6], **Maoxuan Liu**[3], **Huiling Zhang**[1], **Xiaohu Ren**[7], **Yi Pan**[8], **Yin Peng**[9], **Yi Shi**[4], **Xiaochun Wan**[3]*, **Yingxia Liu**[2]*, **Yanjie Wei**[1]*

1 Center for High Performance Computing, Joint Engineering Research Center for Health Big Data Intelligent Analysis Technology, Shenzhen Institutes of Advanced Technology, Chinese Academy of Sciences, Shenzhen, Guangdong, China, 2 Shenzhen Key Laboratory of Pathogen and Immunity, National Clinical Research Center for infectious disease, State Key Discipline of Infectious Disease, Shenzhen Third People's Hospital, Second Hospital Affiliated to Southern University of Science and Technology, Shenzhen, China, 3 Shenzhen Laboratory of Human Antibody Engineering, Institute of Biomedicine and Biotechnology, Shenzhen Institutes of Advanced Technology, Chinese Academy of Sciences, University City of Shenzhen, Shenzhen, China, 4 CAS Key Laboratory of Pathogenic Microbiology and Immunology, Institute of Microbiology, Chinese Academy of Sciences, Beijing, China, 5 School of Biological Sciences, Nanyang Technological University, Singapore, Singapore, 6 University of Chinese Academy of Sciences, Shijingshan District, Beijing, China, 7 Institute of Toxicology, Shenzhen Center for Disease Control and Prevention, Shenzhen, China, 8 Department of Computer Science, Georgia State University, Atlanta, Georgia, United States of America, 9 Department of Pathology, School of Medicine, Shenzhen University, Shenzhen, China

☯ These authors contributed equally to this work.
* xc.wan@siat.ac.cn (XW); yingxialiu@hotmail.com (YL); yj.wei@siat.ac.cn (YW)

**Data Availability Statement:** All relevant data are within the manuscript and its Supporting Information files.

## Abstract

The spread of severe acute respiratory syndrome coronavirus 2 (SARS-CoV-2) virus poses serious threats to the global public health and leads to worldwide crisis. No effective drug or vaccine is readily available. The viral RNA-dependent RNA polymerase (RdRp) is a promising therapeutic target. A hybrid drug screening procedure was proposed and applied to identify potential drug candidates targeting RdRp from 1906 approved drugs. Among the four selected market available drug candidates, Pralatrexate and Azithromycin were confirmed to effectively inhibit SARS-CoV-2 replication *in vitro* with $EC_{50}$ values of 0.008μM and 9.453 μM, respectively. For the first time, our study discovered that Pralatrexate is able to potently inhibit SARS-CoV-2 replication with a stronger inhibitory activity than Remdesivir within the same experimental conditions. The paper demonstrates the feasibility of fast and accurate anti-viral drug screening for inhibitors of SARS-CoV-2 and provides potential therapeutic agents against COVID-19.

## Author summary

Currently, a novel coronavirus called SARS-COV-2 is spreading across many parts of the world. Unfortunately, there is still a lack of effective drugs against the virus. Additionally,

**Funding:** This work was partly supported by the National Key Research and Development Program of China under Grant No. 2018YFB0204403 (Y.W.) and 2019YFA0906100 (X.W.); Strategic Priority CAS Project XDB38000000 to Y.W., National Science and Technology Major Project under Grant No. 2018ZX10101004 (Y.Y.), National Science Foundation of China under Grant no. U1813203 (Y. W.); the National Natural Youth Science Foundation of China (Grant no. 31601028: Y.P.); the Shenzhen Basic Research Fund under Grant no. JCYJ20190807170801656 (J.L.), JCYJ20180507182818013 (Y.W.), JCYJ20170413093358429 (Y.W.), and the SIAT Innovation Program for Excellent Young Researchers (J.L.). The funders had no role in study design, data collection and analysis, decision to publish, or preparation of the manuscript.

**Competing interests:** NO authors have competing interests

it usually takes much longer time to develop a new drug using traditional methods. Thus, it is now better to rely on some alternative methods to develop drugs that can treat such a disease effectively. In this paper, we have proposed a deep learning and molecular dynamics simulation based hybrid drug screening procedure for identifying potential drug candidates targeting RdRp from 1906 market available drugs. Our screening have successfully identified a FDA-approved drug called Pralatrexate that strongly inhibits the replication of 2019-nCoV in vitro with EC50 values of 0.008μM. This work demonstrated the feasibility of accurate virtual drug screening for inhibitors of SARS-CoV-2 and provides potential therapeutic agents against COVID-19.

## Introduction

The Coronavirus Disease 2019 (COVID-19) caused by severe acute respiratory syndrome coronavirus 2 (SARS-CoV-2) has developed into a global pandemic with millions of people infected and tens of thousands of lives lost [1]. To date, no clinically proven drug or vaccine is available. There is an urgent need to identify antiviral agents that can inhibit SARS-CoV-2.

De novo drug development process is time-consuming, unable to meet the urgent need to combat COVID-19. Given current emergencies, repurposing existing approved drugs for COVID-19 may provide a shortcut [2]. Drugs under recent clinical trials such as Remdesivir were shown to inhibit the replication of SARS-CoV-2 *in vitro* [3–5]. The structural basis of the RNA-dependent RNA polymerase (RdRp) inhibited by Remdesivir is well illustrated in a recent work [6]. Some patients have been treated with compassionate-use of Remdesivir and shown significant clinical improvements [7]. As a core component of the RNA synthesis machinery, RdRp is believed to be one of the most promising therapeutic target [8,9]. Molecules that can bind to the catalytic site of RdRp could potentially interfere the viral RNA synthesis [10].

Several computational drug screening methods based on molecular docking, deep learning or Molecular Dynamics (MD) simulations have been applied in drug repositioning studies for COVID-19 [9,11–14]. However, most of these normally rely on a single technique or lacks experimental validation. Each computational method, with its underlying philosophy, often has its own strong and weak points, while a proper combination and modification of such methods could provide a better solution. Previously, we have developed two deep learning-based models to estimate protein-ligand interactions: DFCNN [15] and DeepBindBC (http://cbblab.siat.ac.cn/DeepBindBC/index.php). DFCNN uses molecular vector data of protein pocket and ligand, instead of spatial information at interaction site, to estimate the protein-ligand pair as binding or non-binding with a probability value between 0 and 1. DeepBindBC estimates the binding possibility from atom contact information at interaction surface of a modelled 3D protein-ligand complex. The input of DeepBindBC contains spatial information of the protein-ligand interface, thus it strongly complements DFCNN.

We propose a hybrid screening procedure, based on deep learning and molecular simulation, consisting of DFCNN [15], DeepBindBC, Autodock Vina [16], pocket localized molecular dynamics simulation, and metadynamics, as well as our inhouse developed tools, to explore the binding potential of drugs from TargetMol-Approved_Drug_Library, a drug library containing 1906 of current market available drugs, by TargetMol, for possible repositioning of any drug under current global emergency situation. As a result, four drugs are considered possible interactors of RdRp and selected for further experimental validation, and Pralatrexate is identified as an effective inhibitor of replication of SARS-CoV-2 *in vitro*.

## Results

A hybrid virtual screening procedure is performed, illustrated in Fig 1A. 1906 approved drugs from TargetMol-Approved-Drug-Library were subject to the proposed screening process which consists of molecular vector-based screening, structure-based screening and force field-based screening. DFCNN and DeepBindBC are both deep learning-based methods. Four candidates (Pralatrexate, Azithromycin, Sofosbuvir, Amoxicillin) were selected by the proposed method for experimental validation. The qRT-PCR assay, indirect immunofluorescence assay (IFA) and CCK-8 assay were carried out to validate the efficacy for Pralatrexate, Azithromycin which inhibit SARS-CoV-2 replication *in vitro*. Surface plasmon resonance (SPR) assay was used to evaluate the RdRp-drug binding affinity.

### Deep learning and docking

Interestingly, from S1 Table, we notice that DFCNN, DeepBindBC and Autodock Vina predict different drug compounds as top hits. This indicates that these three methods may be potentially complementary to each other through judging protein-ligand interactions from different perspectives. For instance, Amenamevir and Azithromycin have better Autodock Vina scores whereas Odanacatib and Nitisinone are found to have high DFCNN scores.

We first rejected those drugs that had poor prediction by any of the methods. The 22 drugs with DFCNN score above 0.9 and docking score bellow -7 kcal/mol were firstly selected and presented in S1 Table. Among the 22 drugs, we further excluded the drugs with a DeepBindBC score below 0.7, resulting in 14 drugs shown in bold in S1 Table. Overall there are 5 nucleoside analogues, 2 antibiotic drugs, 3 antivirus drugs, 2 anticancer drugs and 3 other drugs (S1 Fig) selected by the molecular vector-based and structure-based screening process. Sofosbuvir is both a nucleoside analogue and an antivirus drug. These 14 drugs are subject to force field-based screening in the next stage.

Among 2 antibiotic drugs, Azithromycin, a drug used to treat a variety of bacterial infections, showed top Autodock Vina score of -8.6 kcal/mol, good DFCNN score and DeepBindBC score (0.9093 and 0.8589), respectively. Gautret, P. *et al* claimed that combined with Hydroxychloroquine, Azithromycin can have good efficiency in treating COVID-19 with significant viral load reduction [17]. However it should be noticed that currently there is no evidence of the effectiveness of Azithromycin in the treatment of COVID-19 and have many debates about effective of Azithromycin on COVID-19 [18].

The top two predicted molecules by DeepBindBC are nucleotide analogues. Sofosbuvir is a nucleotide analogue inhibitor of hepatitis C virus (HCV) NS5B polymerase to treat infectious liver disease, [19] whereas Clofarabine is a purine nucleoside antimetabolite used for treating refractory acute lymphoblastic leukaemia [20]. More nucleotides analogues in the candidate list were selected by our method, such as Adenosine, Vidarabine, and Gemcitabine, indicating some RdRp-nucleotides interaction patterns have been implicitly recognized by the proposed hybrid drug screening method.

### Force field-based simulation

To further screen the 14 selected drugs (2D structures shown in S1 Fig) and understand their interactions and stability, we have performed MD simulations on RdRp-drug complexes. The structure stability is estimated by Root Mean Square Deviation (RMSD) over 100ns time scales, shown in S2 Fig. The drugs such as Azithromycin, Pralatrexate, Romidepsin, Teriflunomide and Vidarabine are found stable indicated by the minimum RMSD fluctuations. On the other hand, the drugs such as Adenosine, Amenamevir, Fipronil, Gemcitabine and Sofosbuvir have high RMSD fluctuations. The number of hydrogen bonds formed between RdRp and the drug

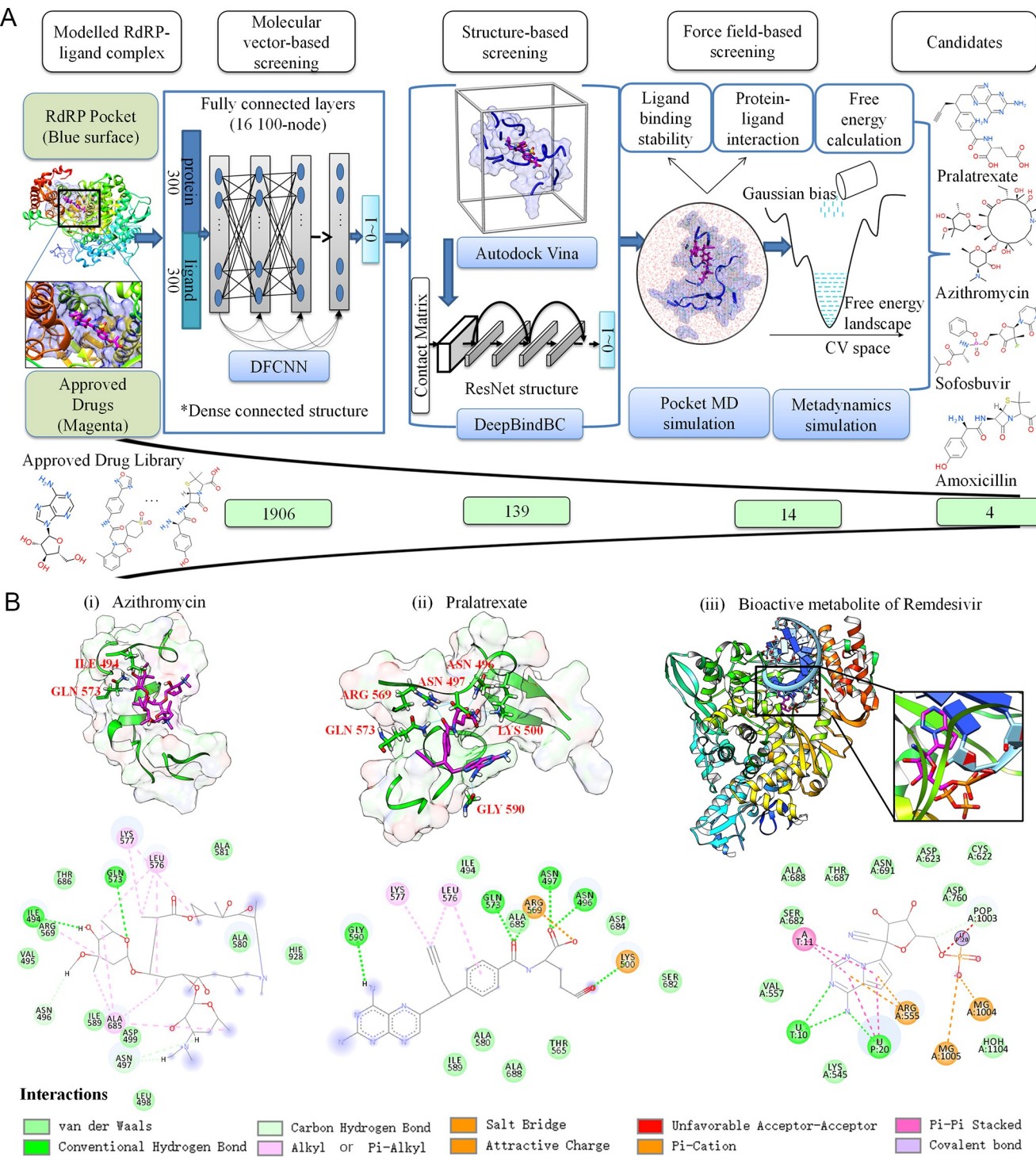

**Fig 1. Drug repurposing against RdRp for COVID-19 using a hybrid deep learning and molecular simulation strategy. A,** 1906 approved drugs were subject to the proposed screening process which consists of molecular vector-based screening, structure-based screening and force field-based screening. DFCNN and DeepBindBC are both deep learning-based methods. 4 candidate drugs were selected by the proposed method, including Pralatrexate, Azithromycin, Sofosbuvir, Amoxicillin. B, Key interactions between the studied drugs and RdRp from the last frame of MD simulation, for (i) Azithromycin and (ii) Pralatrexate. RdRp binding pocket is shown in green with surface representation and the corresponding drugs are shown in magenta. The 2D Schematic diagram of drug-RdRp interaction is given bottom, and neighbor residues (within 4 Å of the drug) are shown. B(iii), the experimental structure of Remdesivir in its monophosphate form with RdRp (PDB ID 7BV2), the 2D Schematic diagram of the interaction was also shown.

in the 100 ns MD simulation was also computed and shown in S3 Fig. Among the 14 RdRp-drug complexes, Pralatrexate clearly showed more hydrogen bonds with the RdRp than others. We found that there is no direct correlation between RMSD fluctuation and size (or number of initial contacts). For instance, the Romidepsin has very large size (molecular weight: 544.7 g/mol), but shows less binding stability compared to Pralatrexate (molecular weight: 479.5 g/mol). Also, Vidarabine have molecular weight of 271.3 g/mol, shows more stable binding compared to Sofosbuvir (molecular weight: 527 g/mol) and Amenamevir (molecular weight: 484.6 g/mol). In MD simulation, small size ligands can also have very tight binding depending on the ligand and binding site (pocket) properties [21]. Whether the binding is stable is more related to the complementary between pocket and ligand in both geometric and physio-chemical features. So the RMSD calculations can help to provide useful information about the binding stability.

The binding free energy *vs* coordination number (CV: collective variable) from metadynamics simulations is shown in S4A Fig. The lowest energy conformations of protein-drug complexes for Amoxicillin, Azithromycin, Pralatrexate and Sofosbuvir showed more contacts in the interface region, as indicated by the high coordination numbers (S4A (iii) Fig), while most other compounds favor smaller coordination number (close to zero) indicating no or weak interactions (S4A (i) Fig).

Detailed interaction patterns between RdRp and the four most optimal compounds (Azithromycin, Pralatrexate, Amoxicillin and Sofosbuvir) are shown in Figs 1B (i)(ii) and S4B (i) (ii), whose structures are taken from the last frame of the 100 ns MD simulation. Azithromycin and Pralatrexate interact with 16 amino acid residues of RdRp to form a stable complex. The RdRp-Azithromycin complex is mainly dominated by van der Waals interactions, whereas Pralatrexate involves more polar and charge interactions. According to the calculated free energy difference (ΔG) values from the metadynamics simulations between the unbound state and the binding state for Amoxicillin, Azithromycin, Pralatrexate and Sofosbuvir (S2 Table), Azithromycin and Pralatrexate (-305.8 kJ/mol, -128.6 kJ/mol) show more favorable binding energy than Amoxicillin and Sofosbuvir (-67.3 kJ/mol and -89.9 kJ/mol). Remdesivir has covalent bond with the RNA primer, our method can only estimate binding free energy between non-colvalent-binding protein and ligand. Furthermore, our system does not contain RNA premier, so the interaction mechanism and pattern of the Pralatrexate and Azithromycin would be different from Remdesivir.

It is noted that all the nucleoside analogues highly recommended by deep learning-based screening methods were excluded from the force field based screening process. The possible explanation is that our protein-drug systems do not contain the RNA primers during the MD simulation, and without RNA the nucleoside analogues would not form base pair like interaction, hence showing no binding in MD simulation.

Interaction patterns (taken from the last frame of 100 ns MD simulation) between RdRp and Azithromycin, Pralatrexate are shown in Fig 1B (i)(ii), respectively. The interaction between Remdesivir in its monophosphate form and RdRp (PDB ID: 7BV2) is also given for comparison in Fig 1B (iii). Azithromycin forms 2 hydrogen bonds with GLN573 and ILE494 through keto and hydroxyl groups respectively, and many hydrophobic related interactions with the RdRp binding site (*e.g.* LYS577, LEU576, ALA685) through alkyl groups, whereas Pralatrexate shows enhanced and more stable interactions with RdRp binding site, including 6 hydrogen bonds with GLN573, ARG569, ASN496, ASN497, LYS500 and GLY590. Pralatrexate also forms Alkyl or Pi-Alkyl interaction with LYS577 and LEU576, and salt bridges with ARG569 and LYS500. Azithromycin and Pralatrexate share 10 common neighbor residues (62.5%) of RdRp, as shown in Fig 1B(i)(ii) and S3 Table, indicating a similar binding cavity.

In a recent study, the authors have shown that the monophosphate active form of Remdesivir interacts with the RdRp and covalently incorporated into partial double stranded RNA

template of RdRp at the +1 position [6]. Shown in Fig 1B (iii) and S3 Table. The key interacting contacts between Remdesivir and RdRp include ARG555. Remdesivir only shares 2 common close contact residues with Pralatrexate, and no common close contact residue with Azithromycin, suggesting a different interaction pattern.

This screening procedure relies on non-covalent bond interactions and assumes RdRp is in apo form (without RNA primer), Fig 1B(i)(ii) shows Azithromycin and Pralatrexate interacting with the region of RdRp which consists of residues such as GLN573, ARG569, ALA685. Comparing our modeled structure with Remdesivir in its monophosphate-RdRp complex, it is possible Azithromycin and Pralatrexate occupy part of the RdRp cavity with the non-covalent binding which may interfere the entry of the RNA primer strand to the cavity.

## Pralatrexate and Azithromycin inhibit the replication of SARS-CoV-2 *in vitro*

To further confirm the efficiency of the hits from the virtual screening, we tested the antiviral activity of Azithromycin, Pralatrexate, Amoxicillin and Sofosbuvir *in vitro*. Experiments were performed in a biosafety level 3 laboratory where regulation requires. Vero cells were infected with SARS-CoV-2 (BetaCoV/Shenzhen/SZTH-003/2020, GISAID No. EPI_ISL_406594) at a MOI(multiplicity of infection, which represents the ratio of the numbers of virus particles to the numbers of the host cells in a given infection medium.) of 0.02 (the cytopathic effect was mild at 48 hours post-infection with this MOI) in the presence of varying concentrations of the tested drugs, and the inhibition rates were evaluated by quantification of viral copy numbers in the cell supernatant via quantitative reverse transcription polymerase chain reaction reverse transcription polymerase chain reaction (qRT-PCR) and confirmed with immunofluorescence assay (Fig 2). The results showed that Pralatrexate and Azithromycin could efficiently inhibit the replication of SARS-CoV-2, with half-maximal effective concentration (EC$_{50}$) values of 0.008 and 9.453 μM (Fig 2A), whereas Remdesivir achieved an inhibitory activity with EC$_{50}$ value of 8.777 μM within the same experimental system (S5 Fig). Indirect immunofluorescence assay (IFA) showed similar results with qRT-PCR assay (Fig 2B). CCK-8 assay of the two drugs showed that the half-cytotoxic concentration (CC$_{50}$) values of Pralatrexate and Azithromycin on Vero cells were 0.167 μM and $>$ 100 μM, respectively, and the calculated the selectivity indexes (SI) of Pralatrexate and Azithromycin were 20.878 and $>$10.579, respectively. Whether the two drugs worked at the stage of viral entry or post entry was analyzed using time-of-addition assay as previously reported [5]. The results showed that Pralatrexate functioned at a stage post virus entry, while Azithromycin functioned at both entry and post-entry stages of the SARS-COV-2 infection in Vero cells (Fig 2C). Furthermore, surface plasmon resonance (SPR) experiments were performed to test the *in vitro* binding of Pralatrexate and Azithromycin with immobilized RdRp protein of SARS-CoV-2. Both drugs showed expected binding response in S6 Fig.

## Discussion

To perform the drug screening process efficiently and accurately is still a challenge for computer-aided drug design. Though recent deep learning-based approaches have demonstrated its potential to be efficient/effective by learning from a sufficient amount of training data, problems such as overfitting, and the discrepancy between training data and real-world data remain [22]. The proposed deep learning and molecular simulation based drug screening method was able to select 4 approved drug candidates targeting RdRp from 1906 drugs, and 2 out of 4 (Pralatrexate and Azithromycin) can effectively inhibit SARS-CoV-2 replication *in vitro* with EC$_{50}$ values of 0.008μM and 9.453 μM. The molecular vector-based deep learning

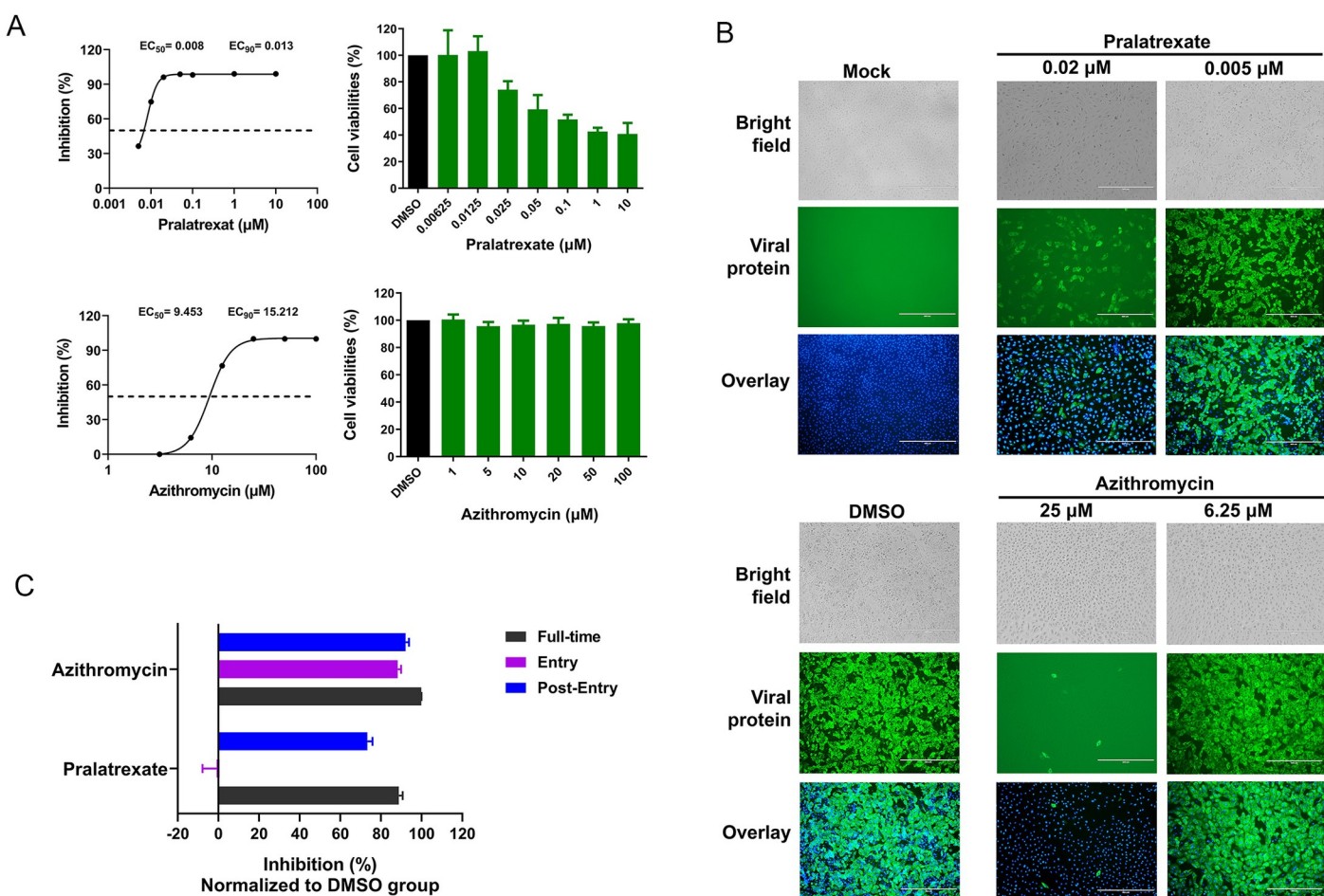

**Fig 2. The antiviral activities of the test drugs against SARS-COV-2 *in vitro*. A**, Vero cells were infected with SARS-COV-2 at an MOI of 0.02 in the presence of the indicated concentrations of the tested drugs for 48 hours. The viral yield in the cell supernatant was then quantified by qRT-PCR. Meanwhile, cytotoxicity of these drugs to Vero cells was measured by CCK-8 assay. **B**, Immunofluorescence microscopy of virus infection upon treatment of Pralatrexate and Azithromycin at the indicated concentrations. IFA was performed at 48 hours post-infection. Scale bar, 100 μm. Cells were immunostained for the Viral protein (green) and DNA (blue). **C**, Time-of-addition experiment of Pralatrexate and Azithromycin. Cells were infected with SARS-COV-2 at an MOI (multiplicity of infection, represents the ratio of the numbers of virus particles to the numbers of the host cells in a given infection medium.) of 0.02 with different treatment, and virus yield in the infected cell supernatants was quantified by qRT-PCR. For the group of "Entry", the drugs were added to the cells for 1 hour before viral attachment, and at 2 hours post-infection, the virus-drug mixture was replaced with fresh culture medium. For the group of "Post-entry", drugs were added at 2 hours post viral infection, and maintained until the end of the experiment. For the group of "Full-time", Vero cells were pre-treated with the drugs for 1 hour, and the virus was then added to allow attachment for 2 hour. Afterwards, the virus-drug mixture was removed, and the cells were cultured with drug-containing medium until the end of the experiment.

method and the structure-based deep learning method are complementary to each other in the sense that high efficiency and accuracy are both achieved. We noticed the experimental activities of Pralatrexate and Azithromycin do not correlate with the free energy of binding. The computational methods of free energy calculations include several limitations. This includes approximations involved for deriving force field parameters, and approximations in estimating energy contributions. An exact correlation between computational calculation and experimental free energy is not possible. Also the experimental activities may not always directly correlate with binding free energy, for instance, weak binding on critical residues may demonstrate better activity than binding on non-critical residues. Nonetheless, our computational calculations show good approximations when used in a screening context.

For the first time, Pralatrexate is found to potently inhibit SARS-CoV-2 replication *in vitro* with a stronger inhibitory activity ($EC_{50}$ value: 0.008μM) than Remdesivir ($P < 0.0001$) under

same experimental setup. Compared with the GHDDI drug list (The Global Health Drug Discovery Institute: https://ghddi-ailab.github.io/Targeting2019-nCoV/preclinical/.) that inhibit SARS-CoV-2 *in vitro*, Pralatrexate showed the smallest $EC_{50}$. Among the 154 current reported drugs by GHDDI, NSC319726 have top inhibitory activity over SARS-CoV-2 ($EC_{50}$ value <0.02µM). Pralatrexate is a folate analogue metabolic inhibitor, which was approved by FDA in 2009 for the treatment of patients with relapsed or refractory peripheral T cell lymphoma (PTCL). Pralatrexate inhibits the folate metabolism pathway through inhibition of dihydrofolate reductase (DHFR) [23]. The peak concentration in plasma (Cmax) can achieve 10.5 µM from a standard dosing regimen [24]. Its Cmax is around 800-fold higher than the $EC_{90}$ of antiviral activity, suggesting a great potential for clinical implications.

Pralatrexate was selected by the virtual screening pipeline based on its potential acts of inhibiting the RNA dependent RNA protease (RdRp) enzyme, whereas, its extremely low EC50 for the virus replication compared to Remdesivir (RdRp inhibitor) may have multiple mechanism of action involved as well. Pralatrexate is known to be an antifolate that efficiently prevents synthesis of DNA and presumably also RNA [25], which may explain inhibition of SARS-CoV-2 replication. Pralatrexate was approved by FDA for patient with terminal disease in spite of its toxicity, therefore, we should be aware that FDA approval does not guarantee the possibility of immediate use of the drug against COVID-19.

Though both Pralatrexate and Azithromycin inhibit SARS-CoV-2 replication *in vitro*, the time-of-addition experiment showed that they functioned at different stages of SARS-CoV-2 infection. Similar to Remdesivir, Pralatrexate mainly inhibited the replication of SARS-CoV-2 at the stages of post-entry. On the other hand, Azithromycin inhibited the replication of SARS-CoV-2 at both entry and post-entry stages like chloroquine [5]. This indicates the Azithromycin may also have multiple mechanism of action.

Out of the 4 selected drug candidates targeting RdRp, Amoxicillin and Sofasbuvir have failed to inhibit SARS-CoV-2 replication *in vitro*. Molecular dynamic simulations show they have deviated from its initial binding position (S4A (i)(ii) Fig) with their ligand RMSD > 1.5 nm for most of the simulation time and large fluctuation was observed (S2 Fig). The calculated free energies difference between binding state and unbound state (coordination number around 0) also indicates better binding for Pralatrexate and Azithromycin than Amoxicillin and Sofasbuvir, shown in S4A (iii) (iv) Fig and S2 Table.

To examine why Sofosbuvir can efficiently inhibit RdRp of hepatitis C virus (HCV) [19] while not RdRp of SARS-CoV-2, we have carried a sequence and structural comparison between RdRp of HCV and RdRp of SARS-CoV-2 virus (S7 Fig). In addition to the low sequence identity (23.75%) between RdRp of HCV and RdRp of SARS-CoV-2 virus, binding pockets of both complexes showed a quite different composition. For instance, there are 5 *vs* 3 ASPs, 2 *vs* 1 LYSs, 1 *vs* 3 GLUs, 0 *vs* 6 ARGs in RdRp pocket of SARS-CoV-2 and RdRp pocket of HCV, respectively. The RdRp pocket of SARS-CoV-2 is more negatively charged, while the RdRp pocket of HCV is more positively charged.

Full system protein-ligand MD simulations for RdRp-Pralatrexate, RdRp-Azithromycin were performed to validate the robustness of the pocket MD method. Compared to pocket MD simulation, similar hydrogen bond numbers as well as similar low RMSD fluctuations in full MD simulation were observed according to S3 and S8A (i)(ii) Figs. Some key neighbor residues in pocket MD simulation for Azithromycin and Pralatrexate were also kept during the full system MD simulation, according to Figs 1B (i)(ii) and S8B (i)(ii). For instance, LEU576, ILE589, ALA580 and ALA685 have formed alkyl related hydrophobic interaction with Azithromycin in the last frames of both simulations, and ARG569, ASN496 and LYS500 of RdRp have formed salt bridge or hydrogen bonds with Pralatrexate in the last frames of both simulations.

The efficiency and effectiveness of the DFCNN method have been examined previously by screening about 10 million drugs targeting 8 representative protein targets taken from the DUD. E diverse data set. The running parameters of DFCNN are same as the current work. DFCNN was able to screen the 10 million drugs within 5 hours using only a workstation with 80 Intel CPU cores (2.00 GHz) and 60 GB RAM. The effectiveness is evaluated by the prediction-random ratio ($Ratio_{0.9}$), shown in S4 Table. For 6 out of 8 protein targets, $Ratio_{0.9}$ is greater than 1.4, indicating DFCNN is able to enrich the active compounds in ten million compound pools. Among the 8 test cases, the DFCNN achieved best performance on HIVPR (Human immunodeficiency virus type 1 protease) with $Ratio_{0.9}$ of ~860 (about 860 times better than random guess in selecting active compounds in terms of TPR). DFCNN performed worse for GPCR proteins (such as CXCR4) and protein with small inner pocket (such as AKT1). The possible reason is that GPCRs have limited number of reliable structure of protein-ligand complexes in our training dataset and membrane proteins may have very different binding mechanism compared to other type proteins. The poor performance for proteins with small inner pocket is likely due to the special physical-chemical and spatial features. As an enzyme, RdRp has large ligand binding cavity and should be suitable for virtual screening by DFCNN.

To study how the molecular vector-based deep learning screening method selects the 139 candidate drugs from 1906 drugs, 1906 drugs were clustered into 20 groups (S9 Fig). Group 20 has the highest ratio of drugs being selected (31/89 drugs, S9A Fig). The drugs in the Groups 19, Groups 20, Group 17 and Group 15 with high selection ratio tend to contain many electron donors and electrical acceptors, likely due to the RdRp pocket containing many charged groups (S10 Fig), including 5 ASPs, 2 LYSs, and 1 GLU. The percentage of charge and polar residues in the RdRp pocket reaches 54.35% (S10 Fig), which explains why DFCNN prefers to select polar and charged drugs for the RdRp. The structure-based screening (Autodock Vina plus DeepBindBC) selected 14 drugs from 139 drugs, 6 drugs belong to Group 15 (S5 Table), including nucleotide analogues as well as Pralatrexate. Groups 17, 18, 19 all have 2 drugs selected after structure-based screening. Belonging to different clusters, Pralatrexate has many hydrogen donors and acceptors while Azithromycin contains a macrocycle, which tends to form hydrogen bond (or salt bridge) and macrocyclic hydrophobic interactions, respectively.

Since Pralatrexate are known to inhibit DHFR, if it also inhibit RdRp as we assume in this paper, DHFR and RdRp binding pocket may show some similarities. To illustrate that, we extract the ligand binding pocket of DHFR (PDB ID 2W9G), and show it in S12 Fig. Comparing with S10B Fig, we can clearly see both pocket contains many charged residues (10 and 8, respectively), and polar residues (14 and 17, respectively), which all shows high percentage over the total residue number and indicates the hydrophilic of the pockets. And some type of residues have very similar number, for instance both pockets contains 5 ASP residues. The highly similar pocket physio-chemical feature may explain that the two pocket have share same inhibitor Pralatrexate.

## Conclusion

Identifying effective drugs that can treat COVID-19 is important and urgent, especially the approved drugs that can be immediately tested in clinical trials. In this work, we have developed a hybrid protocol of combining deep learning methods with molecular simulations to search for potential drug candidates against RdRp that can inhibit the replication of SARS-CoV-2. Four potential drugs were systematically selected for experimental validation, and Pralatrexate and Azithromycin showed an inhibiting effect with $EC_{50}$ values of 0.008μM and 9.453 μM, respectively. Experimental results from qRT-PCR, CCK-8 assay, indirect

immunofluorescence assay (IFA), Time-of-addition and Surface plasmon resonance (SPR) assay show the proposed screening protocol successfully identified a new therapeutic agent Pralatrexate against COVID-19 by targeting RdRp. The hybrid strategy of combining deep learning, molecular docking, MD simulation in a virtual screening pipeline can effectively help with drug repurposing application and facilitate virtual drug screening against other targets in SARS-CoV-2.

## Materials and methods

### Structural modeling of RdRp and drug compound dataset

The RdRp sequence and its modelled structure were obtained from https://zhanglab.ccmb. med.umich.edu/C-I-TASSER/2019-nCov/. The RdRp-ligand model was constructed by I-TASSER [26]. The ligand was taken from the template protein (PDB ID: 3BR9) [27] by COFACTOR algorithm [28] within I-TASSER using structure comparison and protein-protein networks. We extract the amino acids within 1 nm of the ligand as the binding pocket. RMSD between the modeled structure and the recent experimental RdRp structure (PDB ID 6M71) is calculated (~0.516 Å), shown in S11A Fig [10]. RNA polymerase superfamily region is also very similar between these two structures (RMSD = 0.456 Å, S11A Fig).

TargetMol-Approved_Drug_Library, which contains 1906 compounds, was used as virtual screening library. These 1906 compounds collected by TargetMol are drugs approved by Food and Drug Administration (FDA), the European Medicine Agency (EMA), or China Food and Drug Administration (CFDA), or included in the US Pharmacopeia (USP) Dictionary, the British Pharmacopoeia (BP), the European Pharmacopoeia (EP), the Japanese Pharmacopoeia (JP), or Chinese Pharmacopoeia (CP) Dictionary.

### Machine-learning based drug screening

**DFCNN.**    A deep learning-based method, DFCNN (Dense fully Connected Neural Network), has been developed for predicting protein-drug binding probability [15] and used in this paper for the initial drug screening (Fig 1A). DFCNN utilizes the concatenated molecular vector of protein pocket and ligand as input representation, and the molecular vector are generated by Mol2vec [29] which is inspired by the word2vec model in natural language processing. DFCNN model was trained on a dataset extracted from PDBbind database [30]. Negative data samples in the dataset were generated by cross-combination of proteins and ligands from PDBbind database and positive data samples were taken from protein-ligand pairs in experimental structure. The details of the method were described in our previous paper [15], and DFCNN achieved an AUC value around 0.9 for the independent testing set [15]. The model is about ~100,000 times faster than Autodock Vina in predicting protein-ligand binding probability (range 0~1), because it does not rely on the protein-drug complex conformation.

We screen a large scale chemical compound dataset (about 10 million compounds) targeting 8 representative protein targets taken from the DUD.E diverse data set in order to examine the efficiency and effectiveness of the DFCNN method. For each target, the corresponding dataset contains some active compounds (between 40 and 536) in the DUD.E dataset and 10,402, 895 drug-like compounds from ZINC database. The effectiveness is measured by the prediction-random ratio ($Ratio_{0.9}$), defined as $TPR_{0.9}/Random_{0.9}$, where $TPR_{0.9}$ indicates the ratio ($N_{0.9}$/Active_num) between the number of active compounds with a DFCNN score larger than 0.9 ($N_{0.9}$) and the active number of compounds (Active_num). The total number of the compounds (Total_num) with score above 0.9 is defined as NN. The random selection rate ($Random_{0.9}$) is defined as NN/Total_num. Using cutoff score of 0.9, the prediction-random ratio measures the ratio of predicted TPR and random selection TPR.

**DeepbindBC.**   DeepBindBC, an in-house deep learning-based software, is used for structure based drug screening. Unlike the DFCNN, the input of DeepBindBC includes both the physical-chemical information and spatial information between the protein-ligand interfaces (Fig 1A), hence DeepBindBC is able to achieve higher accuracy, but requires protein-drug complex structure information as input generated by Autodock Vina.

Autodock Vina is used to dock the target with the potential ligands [16]. The pocket is determined by the location of ligand in the template protein (PDB ID: 3BR9) [27]. We set the cavity volume space with 3.5 nm, 3.5 nm and 3.5 nm in x, y, z dimensions from the pocket mass center. AutoDock Tools were used to convert PDB file format to PDBQT file format [31]. The exhaustiveness was set to 8; the num_modes was set to 20, and energy_range was set to 3. The scoring function and optimization algorithm of Autodock Vina have been well discussed in a previous article [16]. The TargetMol-Approved_Drug_Library is well prepared by the TargetMol Company in 3D with sdf format. We have prepared the docking ligand structure using standard procedure used in many drug discovery projects. Briefly, the steps are as follows. The ligand was converted from sdf to mol2 by Openbabel, and then converted into pdbqt by scripts in AutoDockTools. Since the Autodock Vina only considers the heavy atoms, and the polar hydrogens, the protonation state before docking is determined by the default methods in AutoDockTools. In this study, we selected the most likely targets for further validation by setting a binding energy threshold value of -7 kcal/mol.

The DeepBindBC is a ResNet model trained over the PDBbind database. In DeepBindBC, the protein-ligand interaction interface information will be converted into figure-like metric, similar to DeepBindRG [22]. By incorporating the cross-docking (docking proteins and ligands from different experimental complexes) conformation as negative training data, DeepBindBC is highly possible to distinguish non-binders. Since DeepBindBC relies on docking conformation and DFCNN only uses molecular vector information, these two methods are complementary to each other and DeepBindBC takes much more time than DFCNN.

## Pocket molecular dynamics and metadynamics

Further drug screening was carried out by force field based molecular dynamic (MD) simulations. The initial protein-drug complexes was from top score conformation Autodock Vina docking, the ligand was edited by pymol software [32] to make it in correct protonation state at pH 7. In this study, we selected 14 drug binding complexes for MD simulation, including Adenosine, Amenamevir, Amoxicillin, Azithromycin, Clofarabine, Fipronil, Gemcitabine, Nitisinone, Pralatrexate, Raltegravir, Romidepsin, Sofosbuvir, Teriflunomide and Vidarabine, respectively.

We also refined a pocket molecular dynamics simulation (pocket MD, S11B Fig) to facilitate the simulation process by only keeping the binding pocket region for simulation. Binding free energy calculation can be estimated by metadynamics simulations to explore whether protein-ligand will bind in solution. Metadynamics relies on addition of a bias potential to sample the free energy landscape along a specific collective variable of interest [33,34]. Note that the binding free energy calculations from Metadynamics may only be suitable for detect the general trend of binding in virtual screening.

The pocket MD is same as the classical MD simulation, except that we only using the pocket region to reduce system size for simulation (S11B Fig), which is inspired by a previous dynamic undocking (DUck) method [35]. An in-house script was used to extract the pocket region of the protein (1nm within the binding ligand), the N terminal and C terminal ends were capped with the ACE and NHE terminals, respectively. We applied a position restrain to the ACE and NHE terminals to maintain the relative conformation of the pocket. MD

simulation was carried out by Gromacs with AMBER-99SB force field [36,37]. The topology of ligand and the partial charges of ligand was generated by ACPYPE [38], which relies on Antechamber [39]. Firstly, we created a dodecahedron box and put the target-ligand complex at the center. A minimum distance from the protein to box edge was set to 1 nm. We filled the dodecahedron box with TIP3P water molecules [40], the counter ions were added to neutralize the total charge using the Gromacs program tool [41]. The long-range electrostatic interactions under the periodic boundary conditions was calculated with Particle Mesh Ewald approach [42]. A cutoff of 14 Å was used for van der Waals non-bonded interactions. Covalent bonds involving hydrogen atoms were constrained by applying the LINCS algorithm [43].

We performed the energy minimization steps with a step-size of 0.001ns, 100 ps simulation with isothermal-isovolumetric ensemble (NVT), and 10ns simulation with isothermal-isobaric ensemble (NPT) for water equilibrium. After that, a 100ns NPT production run (step size 2 fs) was carried out. The Parrinello-Rahman barostat and the modified Berendsen thermostat were used for simulation with a fixed temperature of 308 K and a pressure of 1 atm. RMSD and hydrogen bond number of the trajectory were calculated using Gromacs tools.

The simulation was continued using the metadynamics approach for exploring the free energy landscape. The interface coordination number of atoms of protein ligand complex was used as collective variable (CV). The protein-ligand interface coordination numbers correlate with the numbers of atom contact, and larger coordination number usually indicates that protein-ligand is in binding state.

The coordination number $C$ is defined as follows by Plumed:

$$C = \sum_{i \in A} \sum_{j \in B} S_{ij} \tag{1}$$

and

$$S_{ij} = \frac{1 - \left(\frac{r_{ij}-d_0}{r_0}\right)^n}{1 - \left(\frac{r_{ij}-d_0}{r_0}\right)^m} \tag{2}$$

In the simulation, $n$ was 6, $m$ was 12, $d_0$ was 0 nm and $r_0$ was 0.5 nm. $d_0$ is a parameter of the switching function. $r_{ij}$ is the distance between atom $i$ and atom $j$. The degrees of contacts between two groups of atoms can be estimated by above function(1) [44]. Metadynamics simulation for each protein-ligand system was performed for 100 ns (except protein-Azithromycin, which was extended to 300ns in order to reach the 0 Coordination Number and achieve convergences). During the metadynamics simulation, Gaussian values were deposited every 1 ps with a height of 0.3 kJ/mol. The widths of the Gaussians were 5 for the coordination number. The free energy landscapes of the metadynamics simulations along the CV were generated by the Plumed program and plotted using Gnuplot [45].

## Tools used in analysis

The USCF Chimera, VMD, ICM-browser-Pro (http://www.molsoft.com/icm_browser_pro.html) and Discovery Studio Visualizer 2019 were used to generate the structure and to visualize the 2D protein-ligand interactions [46–48]. Clusfps (https://github.com/kaiwang0112006/clusfps) which depends on RDKit [49] was used to cluster the drugs in the dataset. The drug fingerprint was used as inputs with algorithm of Murtagh [50] being used for clustering 1906 drugs into 20 groups.

## Cell culture

Vero cell (ATCC, CCL-81) was cultured at 37˚C in Dulbecco's modified Eagle's medium (DMEM, Gibco) supplemented with 10% fetal bovine serum (FBS, Gibco) in the atmosphere with 5% $CO_2$. Cells were seeded in 96-well plates and cultured overnight with a density of $5 \times 10^4$ cells/well prior infection or drug feeding. Remdesivir, Azithromycin, Pralatrexat, Sofosbuvir and Amoxicillin were obtained from Selleck Chemicals. All drugs were dissolved in DMSO to prepare 50 mM stock solutions, and stored at -20˚C. DMSO was used in the controls.

## Viral stock titration by 50% tissue culture infective dose ($TCID_{50}$)

$TCID_{50}$ was measured as previously reported [51]. In brief, Vero cells in 96-well plates were grown to 80% confluence and infected with 10-fold serial dilutions of the stock SARS-CoV-2 (BetaCoV/Shenzhen/SZTH-003/2020, GISAID No. EPI_ISL_406594) for 1 h at 37˚C. The inoculum was removed, and cells were overlaid with fresh DMEM plus 2% FBS. At 5 days post infection (d.p.i), plates were assessed for the lowest dilution in which 50% of the wells exhibited cytopathic effects. The values of $TCID_{50}$ were calculated according to the Reed-Muench method [52].

## Evaluation of antiviral activities of the drugs in Vero cells

Firstly, the cytotoxicity of the five drugs on Vero Cells were determined by CCK8 assays (Sangon). Then the antiviral activities of the drugs were evaluated as previously reported with some modification [5]. Vero cells seeded in 96-well plates were pre-treated with the different doses of the indicated drugs for 1 h, and then virus was subsequently added at multiplicity of infection (MOI) of 0.02 to allow infection for 2 h. Then, the virus-drug mixture was removed and cells were further cultured with fresh DMEM with 2% FBS and the indicated concentrations of drugs. At 48 hours post infection (h.p.i), the cell supernatant was collected and viral RNAs were extracted using the QIAamp RNA Viral Kit (Qiagen, Heiden, Germany) for further quantification analysis. The cells were collected for indirect immunofluorescence assay (IFA). All the experiments involving infectious SARS-CoV-2 were handled in BSL-3 facilities at the Shenzhen Third People's Hospital.

## Quantitative reverse transcription polymerase chain reaction

This assay was carried out as described previously [53]. Viral RNAs were extracted from the samples using the QIAamp RNA Viral Kit (Qiagen, Heiden, Germany), and quantitative reverse transcription polymerase chain reaction (qRT-PCR) was performed using a commercial kit (Genrui-bio) targeting the S and N genes. The specimens were considered positive if the Ct value was $\leq$ 38.0, and negative if the results were undetermined. Specimens with a Ct higher than 38 were repeated. The specimen was considered positive if the repeat results were the same as the initial result and between 38 and 40. If the repeat Ct was undetectable, the specimen was considered negative.

## Indirect immunofluorescence assay (IFA)

IFA was carried out as previously reported [54,55]. Vero cells were fixed in 4% formaldehyde at 48 hours post infection. Then cells were permeabilized in 0.5% Triton X-100, blocked in 5% BSA in PBS, and then probed with the plasma of this patient or healthy control at a dilution of 1:500 for 1 h at room temperature. The cells were washed three times with PBS and then incubated with either goat anti-human IgG conjugated with Alexa fluor 488 at a dilution of 1:500

for 1 h (Invitrogen). The cells were then washed and stained with hoechest-33342 (Invitrogen) to detect nuclei. Fluorescence images were obtained and analyzed using EVOS FL Auto Imaging System (Invitrogen).

## Protein expression and purification

The genes for nsp12 of SARS-CoV-2 isolate BetaCov/Wuhan/WH01/2019 (EPI_ISL_406798) was chemically synthesized with codon optimization for insect cells (*Spodoptera frugiperda*) by Synbio Technologies. The sequence was fused with a C-terminal thrombin cleavage site, a 6×His-tag and a 2×Strep-tag, and incorporated into pFastbac-1 plasmid. Recombinant protein was expressed with Hi5 cells at 27°C. Cells were harvested at 48 hpi(hour post infection)and resuspended in 25 mM HEPES pH 7.4, 1 M NaCl, 1 mM $MgCl_2$ and 2mM TCEP. An equal volume of the same buffer supplemented with 0.2% (v/v) Igepal CA-630 (Anatrace) was added and incubated at 4°C for 10 min. Cells were lysed by sonication and the lysate was clarified by ultracentrifugation. Cleared lysates were passed through a 0.22-μm filter film before further purification. The protein was purified by tandem affinity chromatography and SEC.

## Surface plasmon resonance (SPR) assay

The affinities between nsp12 and drugs were measured at room temperature (r.t.) using a Biacore 8K system with CM5 chips (GE Healthcare). The nsp12 protein was immobilized on the chip with a concentration of 100 μg/mL (diluted by 0.1mM NaAc, PH 4.0).

Drug samples were prepared according to procedure 29264621AA of GE Healthcare Life Sciences. 1×PBS solution plus 5% DMSO and 0.005% p20 was used for running and diluting drugs. A blank channel of the chip was used as the negative control. Serial diluted drugs were then flowed through the chip surface. The LMW multi-cycle kinetics was analyzed with the Biacore 8K Evaluation Software (version1.1.1.7442) and fitted with a 1:1 binding model.

## Statistical analysis

Data are presented as the mean ± SD (Standard Deviation). All analyses were performed using GraphPad Prism version 7.0 for Windows (GraphPad Software, San Diego California, USA). Data were subjected to statistical analysis by two-way ANOVA or two-tailed Student's t-test. The P values less than 0.05 were considered statistically significant.

## Supporting information

**S1 Fig. The 2D plot of 14 selected drug candidates.** The 5 compounds in the red box belong to nucleoside analogue; the 2 compounds in the blue box are antibiotic drugs; the 3 compounds in the green box are known antivirus drugs; the 2 compounds within the black box are known anticancer drugs; the 3 compounds in the yellow box are other types.
(TIF)

**S2 Fig. The RMSD plots of 14 selected RdRp-drug complexes for 100ns MD simulation.** The meaning of color boxes is the same as S1 Fig.
(TIF)

**S3 Fig. The hydrogen bond numbers between protein and ligand along the simulation time for the 14 different systems.** The meaning of color boxes is the same as S1 Fig.
(TIF)

**S4 Fig. Force field based screening results. A**, 1D free energy *vs* coordination number as CV (collective variable) from Metadynamics simulations; (i) and (iii), lowest free energy are shifted

to 0 for comparison. (i) 10 drugs have much less interactions with RdRp because their lowest energy basins are near coordination number 0. (iii) The RdRp-drug complex structures in lowest energy basins of 4 drug candidates (Amoxicillin, Azithromycin, Pralatrexate and Sofosbuvir) show coordination number in the range between 400 and 900, indicating many contacts formed between RdRp and the drugs. (ii), lowest free energy for the Azithromycin and Pralatrexate without curve shift. (iv) lowest free energy for the Amoxicillin and Sofosbuvir without curve shift. **B**(i) and (ii) The interaction patterns between Amoxicillin and RdRp, Sofosbuvir with RdRp, respectively. Last frame of MD simulation trajectories were used. Their interaction patterns have deviated from the initial docking conformation according to large RMSD observed in S2B Fig (iii) and (iv), the representative conformations corresponding to lowest energy basin of RdRp-Azithromycin and RdRp-Pralatrexate from metadyanmcis simulations. (TIF)

**S5 Fig. The antiviral activities of Remdesivir against SARS-COV-2 *in vitro*. A**, Vero cells were infected with SARS-COV-2 at an MOI of 0.02 in the presence of the indicated concentrations of Remdesivir for 48 hours. The viral yield in the cell supernatant was then quantified by qRT-PCR. **B**, Immunofluorescence microscopy of virus infection upon treatment of Remdesivir at the indicated concentrations. IFA was performed at 48 hours post infection. Scale bar, 100 μm. Cells were immunostained for the Viral protein (green) and DNA (blue). (TIF)

**S6 Fig. Biophysical interaction profiles of Pralatrexate and Azithromycin to SARS-CoV-2 nsp12 polymerase protein.** SARS-CoV-2 nsp12 polymerase protein was immobilized on the chip and tested for binding with gradient concentrations of candidate compounds. The binding profiles of different drugs are shown in individual panels. Pralatrexate (A); Azithromycin (B). The raw binding curves are shown in the figure. The data shown is a representative result of two independent experiments using different protein preparations. (TIF)

**S7 Fig. Comparing the RdRp sequence and structures between HCV and SARS-CoV-2. A**, Sequence alignment of RdRp of HCV and RdRp of SARS-CoV-2. The low sequence identity (23.75%) between RdRp of HCV and SARS-CoV-2 may explain why Sofosbuvir inhibitor HCV but not RdRp. **B**, Structural superposition of RdRp 3D structures of HCV and SARS-CoV-2 (Left) shows the ligand binding region between RdRp of HCV and SARS-CoV-2 have very different residue composition. For instance there are 5 *vs* 3 ASPs, 2 *vs* 1 LYS, 1 *vs* 3 GLUs, 0 *vs* 6 ARGs in RdRp pocket of SARS-CoV-2 and RdRp pocket of HCV, respectively (Right box). Compared with the pocket of RdRp of HCV, the RdRp pocket of SARS-CoV-2 is more negatively charged. (TIF)

**S8 Fig. Full MD simulation for RdRp-Azithromycin, and RdRp-Pralatrexate complexes. A**, shows the hydrogen bond number and RMSD of ligand along the MD simulations of RdRp-Azithromycin and RdRp-Pralatrexate, respectively ((i) Azithromycin, (ii) Pralatrexate). **B**, the snapshot and interaction mode for RdRp-drugs from 100ns MD simulation ((i) Azithromycin, (ii) Pralatrexate). Drugs are shown as magenta stick. (TIF)

**S9 Fig. An analysis of the molecular vector-based and structure-based screening in selected drugs for RdRp. A**, The 2D structure of representative drugs for the 20 groups of 1906 size dataset. **B**, The number of drugs in each group (red) before molecular vector-based screening, and the number of each group left (blue) after molecular vector-based screening, G20, G19,

G15, G17, G7 have the more drugs left (red in panel a). **C**, the ratio of drugs keeping within each group after molecular vector-based screening (DFCNN).
(TIF)

**S10 Fig. The physico-chemical feature of RdRp pocket by examining the charge and polar residues. A**, superposition of the modeled pocket structure with the experimental pocket structure, the two pockets are highly similar (0.478 Å). **B**, the residues in the modeled pocket structures. The charged residues are showed as green stick, and the polar residues are showed as purple sticks. The name labels of charge and polar residues are given with red color.
(TIF)

**S11 Fig. The modelled RdRp structure and pocket MD procedure. A,** superimposed conformation of modeled RdRp (green) with the experimental obtained RdRp (blue), the modeled ligand are shown in magenta (Left), the DNA/RNA polymerase superfamily region, which was predicted by InterPro (https://www.ebi.ac.uk/interpro/), have high structure similarity between the modeled structure and experimental structure with RMSD of 0.456 Å (Right). **B,** The schematic workflow of the proposed pocket MD simulation. Step 1, Pocket extraction based on 1nm from the known ligand atoms; Step 2, Adding ACE and NHE to the N terminal and C terminal, respectively; Step 3, Adding ligand molecule topology and coordinate; Step 4, Adding water box and counter-ions; Step 5, Restrain the terminal residues, MD simulation and downstream analysis.
(TIF)

**S12 Fig. The physico-chemical feature of DHFR pocket by examining the charge and polar residues. A**, the DHFR protein with its two known binding ligands. **B**, the residues in the pocket which defined as 1 nm distance from the known ligands. The charged residues are showed as green stick, and the polar residues are showed as purple sticks. The name labels of charge and polar residues are given with red color.
(TIF)

**S1 Table. The neighboring residues of Azithromycin, Pralatrexate, and Remdesivir in its monophosphate that shown in Fig 1B are presented.** The common residues between Azithromycin and Pralatrexate are shown in bold, the common residues between Azithromycin and Remdesivir in its monophosphate are marked with "*". The neighbor distance criterion is 4 Å.
(DOCX)

**S2 Table. The list of drugs that have DFCNN score above 0.9, docking score below -7 kcal/ mol, and the corresponding DeepBindBC scores are presented.** The drugs with DeepBindBC scores above 0.7 were indicated in bold fonts.
(DOCX)

**S3 Table. The calculated free energy for four selected drugs based on S4A (ii)(iv) Fig.** The free energy landscapes of other 10 compounds are positive which indicates no binding. Since non-binding drugs are not our interest and hard to estimate exactly binding free energy value, we hasn't list their calculated binding free energy value.
(DOCX)

**S4 Table. 10 million scale drug screening of DFCNN on 8 representative protein targets from the DUD.E diverse data set.**
(DOCX)

**S5 Table. Number of drugs selected after Autodock Vina and DeepBindBC based screening for each Group in S9A Fig.**
(DOCX)

## Acknowledgments

We would like to thank TargetMol for providing the approved drug library. We would like to acknowledge Harvey Wang from Xiamen Novel AIdrug Co., LTD for his valuable advices and help.

## Author Contributions

**Conceptualization:** Haiping Zhang, Yi Pan, Yi Shi, Xiaochun Wan, Yingxia Liu.

**Data curation:** Haiping Zhang, Min Wang, Yin Peng, Yi Shi.

**Formal analysis:** Min Wang, Xiaohu Ren.

**Funding acquisition:** Haiping Zhang, Yang Yang, Junxin Li, Xiaochun Wan, Yanjie Wei.

**Investigation:** Haiping Zhang, Yang Yang, Min Wang, Konda Mani Saravanan, Jinli Wei, Justin Tze-Yang Ng, Md. Tofazzal Hossain, Maoxuan Liu, Huiling Zhang, Xiaohu Ren, Yi Pan, Yi Shi, Yanjie Wei.

**Methodology:** Haiping Zhang, Yang Yang, Justin Tze-Yang Ng, Yi Pan.

**Project administration:** Yang Yang, Xiaochun Wan, Yanjie Wei.

**Resources:** Junxin Li, Md. Tofazzal Hossain, Yi Shi, Yingxia Liu.

**Supervision:** Jinli Wei, Yi Pan, Xiaochun Wan, Yingxia Liu.

**Validation:** Haiping Zhang, Yang Yang, Min Wang, Xiaochun Wan.

**Visualization:** Haiping Zhang, Junxin Li.

**Writing – original draft:** Haiping Zhang, Yang Yang, Min Wang, Konda Mani Saravanan, Justin Tze-Yang Ng, Md. Tofazzal Hossain, Maoxuan Liu.

**Writing – review & editing:** Haiping Zhang, Yang Yang, Junxin Li, Jinli Wei, Huiling Zhang, Yi Pan, Yin Peng, Xiaochun Wan, Yingxia Liu, Yanjie Wei.

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
