## [Decision Letter · Decision Letter 0]

24 Sep 2020

Dear prof Wei,

Thank you very much for submitting your manuscript "A Novel Virtual Screening Procedure Identifies Pralatrexate as Inhibitor of SARS-CoV-2 RdRp and It Reduces Viral Titer in vitro" for consideration at PLOS Computational Biology.

As with all papers reviewed by the journal, your manuscript was reviewed by members of the editorial board and by several independent reviewers. In light of the reviews (below this email), we would like to invite the resubmission of a significantly-revised version that takes into account the reviewers' comments.

We cannot make any decision about publication until we have seen the revised manuscript and your response to the reviewers' comments. Your revised manuscript is also likely to be sent to reviewers for further evaluation.

Sincerely,

Avner Schlessinger

Associate Editor

PLOS Computational Biology

Nir Ben-Tal

Deputy Editor

PLOS Computational Biology

Reviewer's Responses to Questions

**Comments to the Authors:**

Reviewer #1: The authors report an interesting study involving various in silico approaches for drug repositioning and experimental validation of some selected compounds. In my opinion some sentences need some additional explanations. My points are reported below:

Preparation of TargetMol-Approved_Drug_Library in 3D for docking? Protonation state ? 3D structure generator..? no warning / flag alerts for highly toxic compounds ?

When docking with Autodock Vina, the authors keep the best pose or they also investigate some top poses for each ligand as, the best energy pose does not mean that this is true pose in the binding pocket

I do not understand this sentence in the method: clustering 1906 drugs into 20 groups ? There should be much more than 20 groups ? or it depends how the groups are defined.

Could it be that Azithromycin has some good docking score only because it is big ?

Could it be that drugs that are small or making less contacts have high RMSD fluctuations ? and if so, the type of simulation used mainly confirm that a large compound with many polar interactions will stick to a protein while a small compound will be unstable… If so, the tool does not discriminate anything ? Please clarify or comment

I do not understand these sentences:

starting line: 177:

The possible explanation is that our protein-drug systems do not contain the RNA primers

during the MD simulation, and covalent bond formation, such as the Remdesivir in its

monophosphate form, could not be estimated by traditional MD simulation.

Few lines after, if some of the drugs used here are covalent binders, this should be explained further, please optimize the sentence

The authors suggest that Pralatrexate is already known to be active on Cov2 (as seen in the GHDDI list) but that in that list, the molecular mechanism of action of Pralatrexate has never been described or never shown to involve the RdRp target ? Is it correct ? and thus that Pralatrexate not only inhibits DHFR but also RdRp ? If so, is there some similarity between the binding site of both targets ? Please comment

Reviewer #2: The manuscript by Zhang et al. presents very interesting results obtained by applying different in silico methods, followed by reliable experimental validation. Methods are fine and results are well presented. However, some stylistic improvements are needed:

1. In the title: the viral titer is defined as the lowest concentration of virus (number of infectious units / ml) that can still infects cells. Lowering the viral titer actually means "making the virus more virulent", since a lower virus concentration is needed to infect the cell. I suggest changing in "Reduces Viral Replication" or something similar.

2. Lines 69-70: the authors states that "Several computational drug screening methods..." but only cite their own work. Please include some other citation (from a quick googling: https://doi.org/10.1016/j.jiph.2020.06.016 ; https://doi.org/10.1021/acs.jproteome.0c00383 ; https://www.nature.com/articles/s41598-020-70863-9;
https://doi.org/10.1002/minf.202000115 )

3. Fig. 1: please edit figures to make text readable (I'm no more in my 20th, but reading was hard even when wearing glasses...)

4. Lines 118-121: the paragraph (from "Molecular" to "top hits") needs to be rephrased for better clarity.

5. Line 123: "rejected"..."that had poor..."

6. Lines 168-169: van der Waals

7. Line 170: "metadynamics"

8. Lines 173-174: energies in kJ/mol should be rounded to one decimal place

9: Lines 192-194: the paragraph (from "A recent" to "Table 1") needs to be rephrased for better clarity.

10. Line 200: "interacting....which is consist of"

11. Line 202: "it is possible that Azithromycin and Pralatrexate occupy"

12. Line 208: "activity of Azithromycin,..."

13. Line 229: I would not say that the results is "obvious" (maybe "expected")

14. Lines 275-276 "its extremely low EC50"

15. Line 334: "many electron donors and acceptors"

16. Line 367: "within I-TASSER"

17. Lines 394-404: do the reported parameters belong to the current work or are them from ref 11? Please only report relevant information

18. Line 414: "PDB"

19. Line 439: "Note that the..."

20. Lines 445-446: from "Terminals will" to "pocket" needs rephrasing

21: Line 451: "ions were added"

22. Lines 459-460: check spacing

23. Lines 467-468: "indicates that protein-lingand..."

24. Line 474: which "function"?

25. Line 574: "Thanks to TargetMol for providing"

26. Line 829. "The physico-chemical"

Reviewer #3: 1/ In the Force field based simulations 14 compounds are processed. However the free energy of only 4 compounds is given in supp table 3. Why are the other 10 compounds not listed?

2/ It would also be interesting to calculate the free binding energy of Remdesivir because it is also tested in vitro.

3/ Why are only those 4 compounds tested in vitro? what is the rationale to select only those ones?

4/ The experimental activities of the 2 compounds mentioned in the abstract do not correlate with the free energy of binding in supp table 3. Any explanation?

5/ check English + typo's: metadyanmics, ...

6/ DATA AVAILABILITY: Do not write "structure has been deposited in the PDB" but "structure has been retrieved ...". Those structures are the work of other research groups.

**Have all data underlying the figures and results presented in the manuscript been provided?**

Reviewer #1: Yes

Reviewer #2: **No: **Authors declare that all data are fully available, but I didn't find the link to the repository

Reviewer #3: None

PLOS authors have the option to publish the peer review history of their article (what does this mean?). If published, this will include your full peer review and any attached files.

Reviewer #1: No

Reviewer #2: **Yes: **Alessandro Contini

Reviewer #3: No
---

## [Decision Letter · Decision Letter 1]

3 Nov 2020

Dear prof Wei,

We are pleased to inform you that your manuscript 'A Novel Virtual Screening Procedure Identifies Pralatrexate as Inhibitor of SARS-CoV-2 RdRp and It Reduces Viral Replication in vitro' has been provisionally accepted for publication in PLOS Computational Biology.

Before your manuscript can be formally accepted you will need to complete some formatting changes, which you will receive in a follow up email. A member of our team will be in touch with a set of requests. Furthermore, please note the minor stylistic comments made by Reviewer 2.

Best regards,

Avner Schlessinger

Associate Editor

PLOS Computational Biology

Nir Ben-Tal

Deputy Editor

PLOS Computational Biology

Reviewer's Responses to Questions

**Comments to the Authors:**

Reviewer #2: The authors addressed all the reviewers questions and, in my opinion, the article could be accepted. The are some minor stylistic questions, some of them indicated in the attached pdf. I suggest additional text editing before publications.

Reviewer #3: none

**Have all data underlying the figures and results presented in the manuscript been provided?**

Reviewer #2: Yes

Reviewer #3: Yes

PLOS authors have the option to publish the peer review history of their article (what does this mean?). If published, this will include your full peer review and any attached files.

Reviewer #2: **Yes: **Alessandro Contini

Reviewer #3: No

---

## [Editor Report · Acceptance letter]

30 Nov 2020

PCOMPBIOL-D-20-01496R1 

A Novel Virtual Screening Procedure Identifies Pralatrexate as Inhibitor of SARS-CoV-2 RdRp a nd It Reduces Viral Replication *in vitro*

Dear Dr Wei,

I am pleased to inform you that your manuscript has been formally accepted for publication in PLOS Computational Biology. Your manuscript is now with our production department and you will be notified of the publication date in due course.

With kind regards,

Nicola Davies
